# Association between reduced visual-motor integration performance and socioeconomic factors among preschool children in Malaysia: A cross-sectional study

Mohd Izzuddin Hairol [1]*, Naufal Nordin[1], Jacqueline P'ng[1], Sharanjeet Sharanjeet-Kaur[2], Sumithira Narayanasamy[1], Manisah Mohd-Ali[3], Mahadir Ahmad[1], Masne Kadar[2]

1 Faculty of Health Sciences, Centre for Community Health Studies (ReaCH), Universiti Kebangsaan Malaysia, Kuala Lumpur, Malaysia, 2 Faculty of Health Sciences, Centre for Rehabilitation & Special Needs Studies (iCaReRehab), Universiti Kebangsaan Malaysia, Kuala Lumpur, Malaysia, 3 Faculty of Education, Centre of Community Education & Wellbeing, Universiti Kebangsaan Malaysia, Bangi, Selangor, Malaysia

☯ These authors contributed equally to this work.
* izzuddin.hairol@ukm.edu.my

**Data Availability Statement:** All relevant data are within the manuscript and its Supporting information files.

## Abstract

Visual-motor integration (VMI) is related to children's academic performance and school readiness. VMI scores measured using the Beery-Bucktenicka Developmental Test of Visual-Motor Integration (Beery-VMI) can differ due to differences in cultural and socioeconomic backgrounds. This study compared the VMI scores of Malaysian preschoolers with the corresponding US norms and determined the association between their VMI scores and socioeconomic factors. A cross-sectional study was conducted among 435 preschoolers (mean age: 5.95±0.47 years; age range: 5.08–6.83 years) from randomly selected public and private preschools. VMI scores were measured using Beery-VMI in the preschools' classrooms. Information on the socioeconomic characteristics of the preschoolers was obtained using a parent-report questionnaire. One sample t-test was used to compare their VMI scores with the corresponding US norms. Multivariate logistic regression models were used to explore the influence of socioeconomic factors on the preschoolers' VMI scores. Overall, Malaysian preschoolers' VMI performance was similar to the US standardized norms ($p$>0.05). Children from low-income families were twice likely to obtain lower than average VMI scores than those from higher-income families (OR = 2.47, 95%CI 1.05, 5.86). Children enrolled at public preschools were more likely to obtain a lower than average VMI score than those who enrolled at private preschools (OR = 2.60, 95%CI 1.12, 6.06). Children who started preschool at the age of six were more likely to obtain lower than average VMI scores than those who started at an earlier age (OR = 4.66, 95%CI 1.97, 11.04). Low maternal education level was also associated with lower than average VMI score (OR = 2.60, 95%CI 1.12, 6.06). Malaysian preschoolers' Beery-VMI performance compared well to their US counterparts. Some socioeconomic factors were associated with reduced VMI scores. Those from disadvantaged socioeconomic backgrounds are more likely to have reduced

**Funding:** Authors MIH, SSK, SN, MMA, MA and MK received a research grant under the Fundamental Research Grant Scheme from the Ministry of Higher Education, Malaysia (grant number: FRGS/1/2019/SSI09/UKM/02/4). The url of the funder is mygrants.gov.my The funding body had no role in study design, data collection and analysis, decision to publish, or preparation of the manuscript.

**Competing interests:** The authors have declared that no competing interests exist.

VMI performance, potentially adversely affecting their school readiness, cognitive performance, and future academic achievements.

## Introduction

Visual-motor integration (VMI) refers to the ability and extent to which visual perception and motor coordination are well coordinated with each other [1]. Measurement of VMI in children is vital as it is associated with learning-related activities, such as writing [2–4]. Earlier studies reported that children with deficits in VMI had lower academic achievements [5], the inadequate spatial organization of written work [6], and lower performance in mathematics and reading performance [7].

The Beery-Bucktenica Developmental Test of Visual-Motor Integration (Beery-VMI) [1] is commonly used to assess VMI in children with academic-related difficulties. Clinical findings are typically compared with published US norms to determine if a child has poor VMI performance. Hence, VMI scores need to be free from cultural bias to allow correct interpretation of clinical results. Several studies around the world have reported that Beery-VMI performance among preschoolers in South Africa [8,9], Canada [10], and mainland China [11] compared well to the US normative samples [1]. However, Hong Kong and Singaporean preschoolers performed significantly better with Beery-VMI than their age-matched children from the US [12–15]. Contributing factors for these findings include cultural, language, and curriculum differences, indicating that each population should have its local normative VMI performance values. As such, there is a need to ensure that VMI scores for a local population to be accurately measured and compared to the published standardized norms.

Young children's socioeconomic status also influences their learning experience, cognitive development, school readiness, and academic achievement. In the context of early education, the socioeconomic factors include parental education status [16], early education stream [17], household income [18], preschool enrolment age [19], number of school-aged children in the family [20,21], and parental employment status [22]. For example, children who enrolled in preschool at an earlier age had better reading and mathematics skills than children who joined at a later age, with an effect size between 0.15 to 0.30 [19]. In addition, there was a statistically significant positive correlation between maternal education and young children's reading and mathematics skills ($r = 0.44$) [16]. Socioeconomic status also correlates significantly with young children's fine-motor performance, where a significant positive correlation between VMI performance and socioeconomic status ($r = 0.44$) has been reported [23]. A longitudinal study found that South African children with low socioeconomic background had significantly lower VMI scores than their peers from a higher socioeconomic background. When the groups were re-assessed three years later, the differences in their VMI scores persisted [24]. However, for Malaysian preschoolers, the associations between specific socioeconomic factors and their Beery-VMI performance have not been fully explored.

Malaysian children between four to six years old attend preschools before entering mainstream schools at seven years old [17]. Malaysia's multicultural society comprises ethnic Malays, Chinese, Indian, Borneo natives, *Orang Asli* (indigenous people), and other ethnic minorities. Preschools in the country are differentiated based on the medium of instruction (Malay, Chinese, Tamil, or English), curriculum approaches, religious orientation, or political affiliations [25,26]. Preschool education in the country is not mandatory, and its enrolment rate stands at 77% [27]. Therefore, the differences in preschool access and options may

influence the delivery of early childhood education, thereby possibly affecting the VMI performance of the children.

Two studies have reported on Beery-VMI performance among Malaysian children. Samsudin, Abdullah [28] measured VMI performance in children attending public preschools in the north-eastern state of Kelantan. For their relatively small sample size of 4-, 5-, and 6- year old (n = 30 for each age group), the VMI age score was up to 2.28 years lower than the children's chronological age. Another study measured VMI performance for preschool children residing in the southern state of Melaka [29]. The mean Beery-VMI scores for their 359 participants appeared similar to the US normative sample, but these results were not statistically analyzed. They also found that children attending Chinese-stream preschools had significantly higher VMI scores than those attending Malay-stream and Tamil-stream preschools. These findings suggest that VMI performance in young children may be associated with their demographic and cultural backgrounds. Why there are different VMI performances in children residing in different states in Malaysia is currently unknown.

This study's primary aim was to evaluate the influence of various socioeconomic factors on Malaysian preschoolers' VMI performance. As VMI performance can be culture-specific [13,15], we also compared these children's VMI performance with the standardized VMI scores published in the United States [1].

## Materials and methods

### Study participants

Participants were Malaysian citizen preschoolers aged between 60 and 83 months old. This age group represented the children who would attend primary schools for the first time following data collection. Other inclusion criteria for the participants include distance and near visual acuity of 0.3 logarithms of the Minimum Angle of Resolution (logMAR) (equivalent to Snellen 6/9) or better in each eye, and stereoacuity of at least 170 seconds of arc, to ensure that reduced vision did not affect VMI performance [30].

### Sampling of participants

The sample size was calculated based on the number of children born in the state of Selangor and the Federal Territory of Kuala Lumpur in 2014, which was 134,928 (Department of Statistics, Malaysia). Krejcie and Morgan [31] formula, with the desired confidence level of 95% and precision of ±5, was used to calculate a sample size of 380. The dropout rate was estimated at 10%; therefore the final sample size required was 418. A sample size analysis using G*Power 3.1.9.7 [32] indicated that the sample size was able to detect small to medium effects (d = 0.35) with 95% power. Ethics approval was granted by the Universiti Kebangsaan Malaysia's Research Ethics Committee (UKM/PPI/800-1/1/5/JEP-2019-476) and the Ministry of Rural Development, Malaysia (BPAK620-02/01/01 Jld 15). All conducts of this study, described below, followed the tenets of the Declaration of Helsinki.

The study population was clustered by the type of preschool attended. The preschools were (i) KEMAS preschools that catered for children from suburban and rural areas with families of very low income [17] and (ii) private, paid-for-profit preschools. A list of *KEMAS* preschools was obtained from the Selangor and Kuala Lumpur Community Development Departments. A list of registered private preschools was obtained from the Ministry of Education website. Preschools were selected from these lists using simple random sampling. In 2018, approximately 55% of children attended Government-run preschools while the rest attended private preschools [33]. In order to obtain a sample that represented the population, similar percentages of children were sampled from both types of preschools.

The preschools were only approached after research ethics approval was granted, and permission was also sought from the headteachers. Written informed consent was obtained from the children's parents or legal guardians prior to data collection. Assent was also sought from the children. Those who did not assent to or whose parents did not consent were excluded from the study. Children with a self-reported history of visual, neuromuscular, attentional, or behavioral disabilities, obtained from records supplied by parents, were excluded.

### Instruments and procedures

Habitual distance visual acuity was measured for each eye using the LEA Symbols® Pediatric Test Book. Near visual acuity was measured, also for each eye, using the LEA Symbols® Near Vision Card (Good-Lite Co, Elgin, IL). Visual acuity was scored for each correctly recognized symbol and recorded in units of the logarithm of the Minimum Angle of Resolution (logMAR). Stereopsis was assessed using the Frisby Stereotest (Frisby Stereotests, Fulwood, United Kingdom).

The visual-motor integration (VMI) scores were measured using the Beery-VMI Developmental Test (6th edition), excluding the supplemental test. Before the actual administration of the test, one of the investigators, MK, an occupational therapist with more than 15 years of experience in the pediatric field, organized a group training for research team members on Beery-VMI administration. The training used the Beery-VMI (fifth edition) standard guidelines and standardized administration instructions and procedures during data collection to ensure consistency. Investigator NN, a graduate research student, conducted a trial run with 15 children to refine the administration procedures.

All tests were conducted in the preschools' classrooms, with the participating children wearing their existing optical prescription, if any, under sufficient room lighting. For the Beery-VMI test, the children needed to copy the geometric shapes found in the Beery-VMI booklet within 15 minutes following the test's standard instructions. The use of an eraser was prohibited as per the manufacturer's instructions. The Beery-VMI test was administered by NN and assisted by five final year undergraduate Optometry students for visual status measurements. NN scored each booklet according to the degree of accuracy of the drawing using standard measurement equipment and according to the Beery-VMI (fifth edition) scoring criteria. The scoring was supervised by supervisory team member MK, and any discrepancies were agreed upon by consensus. The raw data were converted to standard scores based on the instruction manual to allow comparison between the participants. The standard score for each participant was also categorized according to the Beery-VMI manual.

Demographic and socioeconomic status, including age, gender, ethnicity, preschool type, parents' education, monthly household income, and preschool enrolment age, was obtained using a self-administered questionnaire distributed to the participants' parents or guardians. Parents' education level (maternal and paternal) was categorized into (i) low (up to secondary school) and (ii) high (up to tertiary level). In Malaysia, less-educated workers in the country are defined as employees with secondary school qualifications or lower [34]. Household income was categorized into (i) low income (less than MYR3000 per month) and (ii) MYR3000 and above per month. MYR3000 (USD1 = MYR4.11, as of 21 September 2020) was taken as the cut-off point as it was the median household income for the bottom 40% of household earners in Malaysia (i.e., the B40 group) [35]. The preschool enrolment age was divided into (i) at six years old and (ii) before the age of six years old. The number of children in the family was categorized into (i) 3 or less, and (ii) more than 3. The maternal and paternal employment status was categorized as (i) employed and (ii) unemployed.

## Data analysis

IBM Statistical Package for Social Sciences (SPSS) version 22 was used to analyze the data. All quantitative variables (habitual distance visual acuity, near visual acuity, stereopsis, and VMI standard score) were analyzed using descriptive statistics and given as mean and standard deviation. One sample t-test was used to compare the VMI standard score measured in the sample with the United States VMI norm score (mean = 100, standard deviation = 15).

The VMI standard scores were also categorized, modified from the Beery-VMI 6th edition manual [1]. VMI scores less than 90 were categorized as 'Lower than Average'; VMI scores between 90 and 109 were categorized as 'Average,' and VMI scores higher than 109 were categorized as 'Higher than Average.' Chi-square tests were performed to determine the association between demographic and socioeconomic variables on VMI score categories. The predictor variables that produced a $p$-value of 0.05 or below were included in multivariate logistic regression models. Odds ratios (OR) were determined using logistic regression analyses after adjusting for confounding variables. The OR represents the probability of getting 'Lower than Average' and 'Average' VMI score relative to those with 'Higher than Average' VMI score (i.e., the reference groups).

The first regression model was unadjusted, with the primary independent variables being preschool type, household income, maternal education status, maternal employment status, and preschool enrolment age. The second regression model was adjusted by gender and ethnicity of the participants, as they had been reported to influence preschoolers' VMI scores [9,15]. The significance level, α, was set at 0.05 for all statistical tests.

## Results

### Malaysian sample characteristics

A total of 435 preschool children met the inclusion criteria and completed the Beery-VMI assessment. The mean age of the participants was 5.95±0.47 years. Table 1 summarizes their demographic and socioeconomic distributions.

### VMI standard score for Malaysian preschoolers

Table 2 presents the mean VMI standard score based on the participant's gender and ethnic groups. Overall, the mean VMI standard score for the sample was 100.42 ± 8.58. The VMI score for girls (101.34 ± 8.01) was higher than that for boys (99.57 ± 9.01) and this difference is statistically significant [$t_{(433)}$ = -2.16, p = 0.03]. There were also significant differences in VMI scores between children from different ethnic groups (one-way ANOVA [$F_{(4, 184)}$ = 2.58, p = 0.04]). Tukey's post-hoc test revealed that ethnic Chinese children scored significantly higher than ethnic Malay children ($p$ = 0.02). The mean scores for each ethnicity are within the Average category as defined in the Beery-VMI manual.

### Comparison between Malaysian and US normative samples

The overall participants' standard mean score (100.42 ± 8.58) was not significantly different from the US scores [$t_{(434)}$ = 1.03, p = 0.30]. The participants were also divided into 12 2-month interval age groups based on the Beery-VMI 6th edition manual. The standard mean score for each age group was compared with the US sample's standard score population (mean = 100, SD = 15). Table 3 summarizes these comparisons.

For almost all age groups, the participants' VMI standard scores were not significantly different from the US sample (one-sample t-tests, all $p > 0.05$). However, the 62–63 months scored significantly higher than the US sample (mean difference = 4.67, Cohen's *d* effect

**Table 1. Demographic and socioeconomic distributions of the study participants.**

| Demographic & socio-economic variables | % |
|---|---|
| **Gender** | |
| Male | 52.0 |
| Female | 47.0 |
| **Ethnicity** | |
| Malay | 85.1 |
| Chinese | 6.0 |
| Indian | 2.1 |
| *Orang Asli* (indigenous people) | 4.0 |
| Others | 2.8 |
| **Preschool type** | |
| *KEMAS* (public, government-run) | 55.9 |
| Private (private, independent) | 44.1 |
| **Household income** | |
| < RM3000 per month | 64.3 |
| ≥ RM3000 per month | 35.7 |
| **Maternal education level** | |
| Low (up to secondary school) | 52.6 |
| High (tertiary education) | 47.4 |
| **Paternal education level** | |
| Low (up to secondary school) | 59.7 |
| High (tertiary education) | 40.3 |
| **Preschool enrolment age** | |
| <6 years old | 65.3 |
| 6 years old | 34.7 |
| **Number of children in the family** | |
| 1–3 | 67.9 |
| >3 | 32.1 |
| **Maternal occupation status** | |
| Employed | 66.9 |
| Unemployed | 33.1 |
| **Paternal occupation status** | |
| Employed | 100.0 |
| Unemployed | 0.0 |

**Table 2. VMI mean standard score based on participants' gender and ethnic group.**

| | VMI score (mean ± SD) |
|---|---|
| **Gender** | |
| Boys | 99.57 ± 9.01 |
| Girls | 101.34 ± 8.01 |
| **Ethnicity** | |
| Malay | 99.86 ± 8.52 |
| Chinese | 105.04 ± 6.35 |
| Indian | 102.22 ± 6.67 |
| *Orang Asli* | 100.94 ± 9.67 |
| Others | 102.17 ± 9.68 |

**Table 3. Comparison of the Beery-VMI score between Malaysian and US samples (mean = 100, SD = 15).**

| Age range | n | Mean ± SD | t-value | Confidence interval | | Cohen's d | p |
|---|---|---|---|---|---|---|---|
| | | | | Lower | Upper | | |
| 60–61 months | 8 | 104.00 ± 13.49 | 0.84 | -7.28 | 15.28 | 0.30 | 0.43 |
| 62–63 months | 33 | 104.67 ± 9.87 | 2.72 | 1.17 | 8.17 | 0.47 | 0.01* |
| 64–65 months | 42 | 101.24 ± 9.53 | 0.84 | -1.73 | 4.21 | 0.13 | 0.41 |
| 66–67 months | 40 | 99.05 ± 9.35 | -0.64 | -3.94 | 2.04 | 0.10 | 0.52 |
| 68–69 months | 38 | 100.71 ± 9.00 | 0.49 | -2.25 | 3.67 | 0.08 | 0.63 |
| 70–71 months | 48 | 100.94 ± 8.60 | 0.76 | -1.56 | 3.44 | 0.11 | 0.45 |
| 72–73 months | 83 | 100.83 ± 8.31 | 0.91 | -0.98 | 2.65 | 0.10 | 0.37 |
| 74–75 months | 34 | 101.32 ± 7.49 | 1.03 | -1.29 | 3.94 | 0.18 | 0.31 |
| 76–77 months | 33 | 100.06 ± 7.49 | 0.05 | -2.59 | 2.72 | 0.01 | 0.96 |
| 78–79 months | 32 | 98.09 ± 6.52 | -1.65 | -4.26 | 0.45 | 0.29 | 0.11 |
| 80–81 months | 39 | 97.00 ± 5.97 | -3.14 | -4.94 | -1.06 | 0.50 | 0.003** |
| 82–83 months | 5 | 94.80 ± 8.87 | -1.31 | -16.22 | 5.82 | 0.59 | 0.26 |
| Overall | 435 | 100.42 ± 8.58 | 1.03 | -0.39 | 1.23 | 0.05 | 0.30 |

* significant at p < 0.05

** significant at p < 0.01.

size = 0.43, p = 0.01) while the 80–81 months scored significantly lower (mean difference = 3.00, Cohen's *d* effect size = 0.50, p<0.01).

## Associations between socioeconomic factors and VMI performance

Table 4 shows the VMI standard scores stratified into three categories. A total of 324 (74.5%) of the participants were in the Average group (VMI standard score between 90 to 109). Sixty-eight participants (15.6%) achieved the 'Higher than Average' VMI score of >109. Forty-three participants (9.9%) had 'Lower than Average' VMI score (VMI score <90).

Table 5 shows the distribution of the participants' VMI score categories by socioeconomic characteristics. Participants with Chinese ethnicity had the highest proportion of Higher than Average VMI scores (26.9%), followed by *Orang Asli* children (23.5%). A significantly higher proportion of participants attending *KEMAS* preschool (13.4%) and from low-income families (12.5%) had Lower than Average VMI scores compared to those attending private preschools (5.9%) and from higher-income families (5.5%) (both p = 0.03). A significantly higher proportion of participants who started preschool education at six years old (a year before formal schooling started) had Lower than Average VMI score (19.3%) compared to those who started their preschool education at the age of five years old or earlier (5.3%) (p<0.001). Low maternal education level was associated with a Lower than Average VMI score (12.3%) compared to high maternal education level (6.8%) (p = 0.08). There was no significant association between

**Table 4. Distribution of VMI standard scores based on three categories: Lower than average, average, and higher than average.**

| VMI standard score category | n | Percent |
|---|---|---|
| Lower than average (<90) | 43 | 9.9 |
| Average (90–109) | 324 | 74.5 |
| Higher than average (>109) | 68 | 15.6 |

**Table 5. Distribution of mean VMI standard scores, percentages, $X^2$- and $p$-values by demographic and socioeconomic characteristics.**

| | VMI standard score category | | | | | | $X^2$ | $p$-value |
|---|---|---|---|---|---|---|---|---|
| | Lower than Average (<90) | | Average (90–109) | | Higher than Average (>109) | | | |
| | Mean ± SD | % | Mean ± SD. | % | Mean ± SD. | % | | |
| **Gender** | | | | | | | | |
| Boys | 83.52±4.99 | 11.9 | 99.55±5.24 | 73.5 | 112.79±3.27 | 14.6 | 2.41 | 0.30 |
| Girls | 83.81±4.81 | 7.7 | 100.70±4.83 | 75.6 | 112.26±2.36 | 16.7 | | |
| **Ethnicity** | | | | | | | | |
| Malay | 83.69±4.58 | 10.7 | 99.77±5.13 | 74.9 | 112.19±2.26 | 14.5 | 7.33 | 0.50 |
| Chinese | - | 0.0 | 102.05±4.33 | 73.1 | 113.14±2.61 | 26.9 | | |
| Indian | 88±0.00 | 11.1 | 103.14±3.81 | 77.8 | 110±0.00 | 11.1 | | |
| *Orang Asli* | 71.00±00 | 5.9 | 100.00±3.79 | 70.6 | 111.25±1.50 | 23.5 | | |
| Others | 86.50±0.71 | 16.7 | 103.78±5.40 | 75.0 | 119±0.00 | 8.3 | | |
| **Preschool type** | | | | | | | | |
| *KEMAS* | 82.77±5.37 | 13.4 | 99.62±4.88 | 72.8 | 112.66±2.62 | 13.8 | 7.33 | 0.03 |
| Private | 85.83±2.13 | 5.9 | 100.65±5.24 | 76.4 | 112.39±3.03 | 17.7 | | |
| **Household income** | | | | | | | | |
| <MYR3000 | 83.03±5.30 | 12.5 | 99.86±5.02 | 74.1 | 112.49±2.48 | 13.3 | 7.26 | 0.03 |
| ≥MYR3000 | 85.75±2.44 | 5.5 | 100.85±5.08 | 74.7 | 112.07±2.37 | 19.9 | | |
| **Maternal education level** | | | | | | | | |
| Low | 83.77±4.83 | 12.3 | 99.59±5.08 | 74.4 | 112.61±2.38 | 13.3 | 5.11 | 0.08 |
| High | 83.00±5.67 | 6.8 | 100.89±5.08 | 74.2 | 112.06±2.46 | 18.9 | | |
| **Paternal education level** | | | | | | | | |
| Low | 82.71±4.91 | 11.9 | 99.81±5.07 | 73.3 | 112.29±2.33 | 14.8 | 2.87 | 0.24 |
| High | 85.36±4.88 | 6.9 | 100.67±5.04 | 75.5 | 112.36±2.60 | 17.6 | | |
| **Preschool enrolment age** | | | | | | | | |
| 6 years old | 82.44±5.55 | 19.3 | 98.50±5.01 | 68.6 | 112.59±3.08 | 12.1 | 20.33 | < 0.001 |
| ≤5 years old | 85.93±2.30 | 5.3 | 100.98±4.94 | 76.9 | 112.19±3.08 | 17.8 | | |
| **Maternal employment status** | | | | | | | | |
| Employed | 85.07±3.62 | 10.0 | 100.19±5.04 | 72.9 | 112.17±2.41 | 17.1 | 1.06 | 0.59 |
| Unemployed | 81.25±5.93 | 9.0 | 100.23±5.26 | 77.4 | 112.61±2.50 | 13.5 | | |
| **Paternal employment status** | | | | | | | | |
| Employed | 83.63±4.87 | 9.89 | 100.11±5.07 | 74.48 | 112.51±2.83 | 15.63 | - | - |
| Unemployed | - | 0.0 | - | 0.0 | - | 0.0 | | |
| **Number of children in the family** | | | | | | | | |
| 1–3 | 83.67±4.61 | 10.9 | 100.38±5.10 | 72.4 | 112.33±2.31 | 16.8 | 2.41 | 0.30 |
| >3 | 83.50±5.89 | 7.6 | 99.55±4.99 | 79.4 | 113.06±4.04 | 13.0 | | |

paternal education level, paternal employment status, maternal employment status, and the number of children in the family with VMI score categories.

Table 6 shows the odds ratio (OR) of the logistic regression models. The first model was unadjusted, while the second regression model was adjusted for gender and ethnicity. Children enrolled in *KEMAS* preschools were about two times more likely to obtain a Lower than Average VMI standard score [OR = 2.47 (95% CI 1.05, 5.86)] compared to those who enrolled in private preschools. Those from low-income families (<MYR3000) had more than three times the odds of having a Lower than Average VMI standard score [OR = 3.70 (95% CI 1.45, 9.39)], compared to those from higher-income families (≥MYR3000). Children whose mothers had up to secondary school education only were 2.60 times more likely to have Lower than Average

**Table 6. Odds ratio for VMI performance by socioeconomic variables.**

| Socio-economic variable | VMI score category | Unadjusted model[a] | | Adjusted model[a] | |
|---|---|---|---|---|---|
| | | OR | 95% CI | OR | 95% CI |
| Preschool type[b]<br>• *KEMAS* (public preschool) | Lower than Average | **2.91**[*] | 1.28, 6.59 | **2.47**[*] | 1.05, 5.86 |
| | Average | 1.23 | 0.73, 2.07 | 1.72 | 0.78, 3.82 |
| | Higher than average | 1 | - | 1 | - |
| Household income[c]<br>• Low (<MYR3000) | Lower than Average | **3.42**[*] | 1.37, 8.54 | **3.70**[*] | 1.45, 9.39 |
| | Average | 1.48 | 0.86, 2.56 | 1.57 | 0.90, 2.75 |
| | Higher than average | 1 | - | 1 | - |
| Maternal education level[d]<br>• Low (up to high school) | Lower than Average | **2.57**[*] | 1.12, 5.89 | **2.60**[*] | 1.12, 6.06 |
| | Average | 1.43 | 0.83, 2.47 | 1.47 | 0.84, 2.56 |
| | Higher than average | 1 | - | 1 | - |
| Preschool enrolment age[e]<br>• 6 years old | Lower than Average | **5.33**[*] | 2.28, 12.49 | **4.66**[**] | 1.97, 11.04 |
| | Average | 1.31 | 0.71, 2.40 | 1.24 | 0.67, 2.29 |
| | Higher than average | 1 | - | 1 | - |

[*] OR is significant at $p<0.05$.

[**] OR is significant at $p<0.01$.

[a]The reference category for VMI score is Higher than Average. Gender and ethnicity factors are considered in the adjusted model.

[b]The reference category for preschool type is private preschool.

[c]The reference category for household income is income above MYR3000.

[d]The reference category for maternal education level is high (tertiary level).

[e]The reference category for preschool enrolment age is less than 6 years old.

VMI standard score [OR = 2.60 (95% CI 1.12, 6.06)] than those whose mothers had tertiary education. Children who enrolled in preschool at the age of six (the year before primary schooling starts) had more than four times the risk of having a Lower than Average VMI standard score [OR = 4.66 (95% CI 1.97, 11.04)] relative to those who enrolled into preschool at an earlier age.

## Discussion

Overall, the mean VMI standard score for Malaysian preschoolers was similar to the standardized US norms. However, those aged 62–63 months, one of the youngest age groups, had significantly better VMI scores than the standardized norms. These could partly be due to their early enrolment into preschool education, as children who attend a preschool program earlier than their peers perform better on various academic-related assessments [19]. We also found that older children (those within the age range between 78 to 83 months) had lower VMI scores than the US standardized scores. On a closer inspection, children in the older age group were mainly from the low household income (77.5%) and enrolled in the government KEMAS preschools (77.1%). The combination of these factors in the older children could have led to late exposure to activities that involve fine-motor skills, which are part of learning in preschools might explain why these older children tend to have lower Beery-VMI scores. Although the differences in the mean scores between these two specific age groups (62–63 months and 80–81 months) and the US standardized scores are statistically significant, they are not clinically significant as the scores are still within the Average category. Therefore, we can conclude that our sample of Malaysian preschoolers' overall Beery-VMI performance compares well with the standard US sample.

We also found that girls had higher Beery-VMI scores than boys, similar to the findings of an earlier study where girls obtained higher scores in Beery-VMI's visual perception subtest [8]. Although the mean VMI score for girls was statistically higher than that for boys in our study, the difference is not clinically significant as their VMI scores are still within the Average category. The absence of an apparent gender bias indicates that the Beery-VMI is a suitable test for measuring VMI performance in our Malaysian sample. In terms of ethnicity, children with Chinese ethnicity had the highest mean VMI score than other ethnic groups. These findings mirror those of a previous Malaysian study, also conducted on multi-ethnic samples of preschoolers, that Chinese-stream preschoolers had the highest mean VMI score compared to Malay-stream and Tamil-stream preschoolers [29]. Although ethnic Chinese children in our study had significantly higher mean VMI scores, the mean VMI scores for all ethnic groups fell within the Average category. Thus, our study confirms that the Beery-VMI, whose scores are standardized in the US, may be used to analyze VMI-related functional difficulties in Malaysian children of different ethnicities.

Interestingly, *Orang Asli* (indigenous) children in this study had the second-highest mean VMI score. This finding was somewhat unexpected, as *Orang Asli* in Malaysia generally faces poverty, poor nutritional health status, and poor cognitive performance [36,37]. However, as this study was conducted in Malaysia's urbanized regions, the *Orang Asli* children would have had better access and opportunities to preschool education, along with various government programs aimed at improving *Orang Asli*'s quality of life and well-being [36]. Our findings also indirectly highlight the importance of access to early education for all children, including those with disadvantaged backgrounds, to ensure timely development of learning-related skills such as the VMI. *Orang Asli* children who reside in more remote areas would likely have different VMI performance.

Our findings on VMI performance for multi-ethnic Malaysian children contrasted with those reported in neighboring Singapore [15]. Singaporean preschool children obtained higher VMI scores than their US peers, particularly those with Chinese ethnicity [15]. Ethnic Chinese children in Hong Kong also had significantly higher VMI scores than the US norms [13,14]. These findings were attributed partly to particular practices in Chinese cultures, such as the writing of Chinese characters, that may have facilitated their VMI performance. Another interesting finding by Lim et al. [15] is that the mean VMI scores for children from each ethnic group in Singapore (Chinese, Indian, and Malay) were significantly higher than the US mean VMI score. The development of Singaporean children's learning-related skills, such as the VMI, may be influenced by Singapore's more formalized, academic-type preschool curriculum in preparation for the demands of the primary school system [38]. Preschool education in the US, on the other hand, emphasizes children's individualism, independence, creativity, and liberty [39,40], including being socially and emotionally positive individuals [41]. The differences in country-specific preschool curriculum focus could be part of the contributing factors that influence young children's VMI performance.

This study's results have also demonstrated the associations between various socioeconomic variables on preschoolers' VMI performance. Many researchers regarded socioeconomic status as one of the most influential factors that can have direct negative impacts on the development of young children [23,42–44]. We found that a higher proportion of children enrolled in *KEMAS* (public) preschools had reduced VMI performance than those enrolled at private, fee-paying preschools. This finding could be due to differences in curriculum and teaching & learning approaches between the two preschool types, where Malaysian public preschools emphasize social and emotional development, while Malaysian private preschools emphasize cognitive development [17]. A recent study reported that Malaysian preschoolers who followed the international Montessori Curriculum performed better in their cognitive, social,

and language skills than their peers who followed the Malaysia National Preschool Curriculum [45]. Public and private preschools in Malaysia also differ in several other aspects. In Malaysian public preschools, the typical teacher-to-child ratio is 1:25, while in private preschools, it is 1:15. Teaching and learning in public preschools are teacher-centered, while in private preschools, it is child-centered [17]. Thus, there is a need to improve the quality of preschool education delivery in the country. A high-quality preschool education workforce and delivery will help improve a child's early education experience, thus prepare them for a smooth transition to formal education in primary schools.

Our study found that children from families with low household incomes were more likely to have reduced VMI performance. Earlier studies have reported that preschoolers with lower economic status were more likely to have poorer fine-motor and VMI performance than their peers with high socioeconomic status [8,44,46]. In an earlier study, Malaysian children from the north-eastern state of Kelantan, a state with the lowest gross domestic product (GDP) per capita in Malaysia [35], had reduced VMI performance [28]. Meanwhile, children from the southern state of Melaka, a state with GDP per capita 3.5 times higher than that for Kelantan, had VMI performance comparable to the US norms [29]. Melaka also has a gross household income that is 1.7 times higher than that of Kelantan [47]. The economic differences at the state and household levels may partly explain the different VMI performance of the children in the two states. Besides, having a low income is associated with parenting stress and responsiveness, which in turn, adversely associates children's cognitive development and behavior [48,49]. Unsurprisingly, Malaysian children with lower socioeconomic status are more likely to have poorer IQ [50] and lower academic achievements [21] and are also likely to have reduced VMI performance. However, whether reduced VMI performance is associated with preschool children's academic performance remains to be investigated. Currently, there are no measurements of academic performance in Malaysia's National Preschool Curriculum [51]. Alternatively, the relationship between VMI score and preschool children's cognitive performance could be explored in future research.

We also found that preschool enrolment age was significantly associated with preschoolers' VMI performance, where children who started preschool education at the age of six (a year before a majority of Malaysian children start their formal schooling) had lower VMI performance compared to those who started at the age of five and younger. The public *KEMAS* preschools prioritize enrolment for 5- and 6-year-old children [52]. Therefore, younger children could not start their preschool education earlier unless their parents can afford the fees charged by private preschools that accept children as young as three years old. Factors such as more time spent in preschool, enrolment at the age of three, or attending more hours per week were associated with substantial developmental gains [53] and better achievement in learning new academic skills and vocabularies [19]. Children who enrolled in an early education program and center-based child care are also less likely to come from economically disadvantaged families [19]. Differences in preschool education opportunities may contribute to gaps in a child's development and widen early childhood education equity [54]. For this reason, there is a pressing need for policymakers in the country to address this issue to ensure fair and timely access to preschool education for all children.

We found that paternal education status was not associated with the VMI performance in preschool children. It has been reported that improvements in children's educational outcomes are associated with the amount of father-child time rather than the father's educational level [55]. However, low maternal education status was found to be associated with reduced VMI performance in preschool children. Our results are in line with the findings of other studies that children whose parents had higher education levels were more likely to have better cognitive performance [50], higher academic achievements [21], and lower social deficiencies

[56]. Maternal practices, such as more convenient access to information, may help with children's health, social, and emotional well-being, all of which feed into their children's literacy development [57]. As women, including mothers, are a crucial part of Malaysia's labor force, it is imperative that they are supported by strengthening child care services, thus ensuring timely and smooth development of their children's overall well-being.

However, we did not find that maternal employment status to be significantly associated with the children's VMI performance. Although employment status, together with educational status and household income are contributors to the well-being and development of young children [58], the effects of maternal employment on child outcomes are inconclusive, as employment does not necessarily cause reduced mother-child time and not all types of parental time directly benefit child development [59]. In addition, all of the children involved in this study had fathers who were formally employed. Therefore, paternal employment status was not included as a socioeconomic variable in our analysis. This finding is unsurprising, as Malaysian societies are still, at large, divide their roles based on gender, where men are primarily expected to provide for the family economically. In comparison, women are more likely to be out of employment due to childcare duties [60].

One of the limitations of this study is that the sample was biased towards children with Malay ethnicity and did not fully reflect Malaysia's ethnic composition. The public *KEMAS* preschools are typically attended by Malay children, especially those who reside away from city centers. This study's findings also only reflect the VMI performance of Malaysian children residing within Klang Valley, an area with the highest GDP per capita in the country. An earlier study has reported that reduced VMI performance was more prevalent among preschoolers in Malaysia's rural parts [28]. Furthermore, all of the socioeconomic variables, including household income and parental employment status, were dichotomous. Although the country's context supported the binary categorization of these variables, the dichotomy would have led to reduced statistical power to detect relationships between the variables. Nevertheless, this study's findings suggest that the Beery-VMI US norms may be used by local clinicians when making interpretations of the Beery-VMI score. This study also highlights the need for children with VMI difficulties to undergo occupational therapy intervention programs to improve their handwriting skills [61,62], thus easing their transition into formal schooling.

## Conclusions

The VMI performance of Malaysian preschoolers is comparable to the US standardized norms. It is also adversely associated with various socioeconomic factors. Young children from disadvantaged socioeconomic backgrounds (from families with low household income, enrolled in rural public preschools and at the year before primary schooling, and low maternal education status) are at a higher risk of having reduced VMI performance. Timely interventions should be initiated as reduced VMI performance in these children may affect their cognitive performance, school readiness, and future academic achievements.

## Supporting information

**S1 Dataset.**
(XLSX)

**S1 File.**
(DOCX)

**S2 File.**
(DOCX)

## Acknowledgments

The authors would like to thank the principals of private preschools where the data collection was conducted, and the Ministry of Rural Affairs, Malaysia, for their co-operation during the conduct of the research project, and to Dr. Baharuddin Haji Omar for his valuable comments on the early stages of the manuscript. Special thanks to Ng Lik Yong, Nurlin Erlina Abdul Manap, Aidil Fadzly Mohammad Zahid, and Che Nursyafika Nazira Suhaidin who contributed with data collection.

## Author Contributions

**Conceptualization:** Mohd Izzuddin Hairol, Naufal Nordin, Sharanjeet Sharanjeet-Kaur, Sumithira Narayanasamy, Manisah Mohd-Ali, Mahadir Ahmad, Masne Kadar.

**Data curation:** Naufal Nordin, Jacqueline P'ng, Masne Kadar.

**Formal analysis:** Mohd Izzuddin Hairol, Naufal Nordin, Jacqueline P'ng, Sharanjeet Sharanjeet-Kaur, Manisah Mohd-Ali, Mahadir Ahmad, Masne Kadar.

**Funding acquisition:** Mohd Izzuddin Hairol, Sharanjeet Sharanjeet-Kaur, Sumithira Narayanasamy, Manisah Mohd-Ali, Mahadir Ahmad, Masne Kadar.

**Investigation:** Naufal Nordin, Jacqueline P'ng.

**Methodology:** Mohd Izzuddin Hairol, Sharanjeet Sharanjeet-Kaur, Sumithira Narayanasamy, Mahadir Ahmad, Masne Kadar.

**Project administration:** Mohd Izzuddin Hairol, Sharanjeet Sharanjeet-Kaur, Sumithira Narayanasamy, Manisah Mohd-Ali, Mahadir Ahmad, Masne Kadar.

**Resources:** Mohd Izzuddin Hairol.

**Supervision:** Mohd Izzuddin Hairol, Sharanjeet Sharanjeet-Kaur, Sumithira Narayanasamy, Manisah Mohd-Ali, Mahadir Ahmad, Masne Kadar.

**Validation:** Mohd Izzuddin Hairol, Masne Kadar.

**Visualization:** Mohd Izzuddin Hairol, Naufal Nordin.

**Writing – original draft:** Naufal Nordin, Jacqueline P'ng.

**Writing – review & editing:** Mohd Izzuddin Hairol, Sharanjeet Sharanjeet-Kaur, Masne Kadar.

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
