## [Decision Letter · Decision Letter 0]

10 Dec 2020

PONE-D-20-32756

Association between reduced visual-motor integration performance  and socioeconomic factors among preschool children in Malaysia: a cross-sectional study

PLOS ONE

Dear Dr. Hairol,

Thank you for submitting your manuscript to PLOS ONE. After careful consideration, we feel that it has merit but does not fully meet PLOS ONE’s publication criteria as it currently stands. Therefore, we invite you to submit a revised version of the manuscript that addresses the points raised during the review process.

The reviewers both find this study to be well-written with clearly presented data. However, a clear rationale for the study is needed. 

We look forward to receiving your revised manuscript.

Kind regards,

Krista Kelly, Ph.D.

Academic Editor

PLOS ONE

Additional Editor Comments:

1. The authors should clearly state how this study differs from the previous literature in the same population. It appears that the authors are trying to make the case that their study is evaluating more socioeconomic factors than previous studies. If so, this needs to be explicitly stated and should be the primary goal of their study rather than the comparison to US norms which has already been done in a previous study.

2. The authors switch between the 5th and 6th edition of the Beery-VMI test (fifth in training, but 6th for testing; 5th for scoring the drawing;6th to standardize scores....) Is this a mistake? If not, please provide rationale for using different editions for various aspects of the study. How much do they differ?

3. Why was the father's occupation not taken into account, but the mother's was?

4. When comparing scores between different ethnic groups, and with the 2 month groups (table 3), did the others control for multiple comparisons (i.e. bonferroni correct?).

Journal Requirements:

2. Please include additional information regarding the survey or questionnaire used in the study and ensure that you have provided sufficient details that others could replicate the analyses. For instance, if you developed a questionnaire as part of this study and it is not under a copyright more restrictive than CC-BY, please include a copy, in both the original language and English, as Supporting Information, or include a citation if it has been published previously.

3. In statistical methods, please clarify whether you corrected for multiple comparisons.

4. In your statistical analyses, please state whether you accounted for clustering by school. For example, did you consider using multilevel models?

5. In your discussions and conclusions please take care to avoid statements implying causality from correlational research. For example, avoid the use of terms such as "predictors/ predictions" or “effects” or “resulted in." Instead consistently use terms such as "associated with" or "associations."

6. We note that you have indicated that data from this study are available upon request. PLOS only allows data to be available upon request if there are legal or ethical restrictions on sharing data publicly. For information on unacceptable data access restrictions, please see http://journals.plos.org/plosone/s/data-availability#loc-unacceptable-data-access-restrictions.

Reviewers' comments:

Reviewer's Responses to Questions

**Comments to the Author**

1. Is the manuscript technically sound, and do the data support the conclusions?

Reviewer #1: Yes

Reviewer #2: Partly

2. Has the statistical analysis been performed appropriately and rigorously? 

Reviewer #1: Yes

Reviewer #2: Yes

3. Have the authors made all data underlying the findings in their manuscript fully available?

Reviewer #1: Yes

Reviewer #2: Yes

4. Is the manuscript presented in an intelligible fashion and written in standard English?

Reviewer #1: Yes

Reviewer #2: Yes

5. Review Comments to the Author

Reviewer #1: The authors examined visuomotor integration in a large cohort of pre-school Malaysian children, compared their performance to US standardized norms, and assessed the influence of SES on VMI scores. Results showed that the Beery-VMI is a suitable tool to evaluate visuomotor integration in Malaysian preschoolers of various ethnic backgrounds. Importantly, the study found that children from lower income families and those that begin preschool later are more likely to perform poorly on the VMI test. These findings have implication for educational policy Malaysia, and also offer broader insights about the role of SES in child’s development. The manuscript is well written, and the methodology is appropriate. I have only a few minor comments for authors to consider.

Minor Comments:

1. Abstract: provide the age range, and consider changing months to years (also in the methods section)

2. Intro, pg 4 (last paragraph): Children who were enrolled…. In the same paragraph the authors should add information about the effect size that was found in the reviewed studies. For example, what was the % difference between groups or the size of correlation that was mentioned?

3. Pg 5 (last paragraph): “as a whole, Beery VMI scores …” – are the authors referring to the composite (total) score?

4. Methods: the age range reported here is 5 to 6.9 yrs, are all these children starting grade 1 at the same time? Ie, are some children starting at 6 and others at 7 years?

5. It’s unclear what effects were used to calculate the sample size. The authors mention that they need 380 children to detect something with a confidence level of 95%, but what is the effect they were trying to detect?

6. Pg 16 (first paragraph): “Table 6 shows that the odds…”. “model was adjusted for gender…”

Reviewer #2: Overall it is a well written paper with clearly identified objectives and appropriate analysis. The author highlighted a need in general to evaluate VMI within differing geographical populations due to previous findings, however they didn’t identify the need for it specifically to be evaluated in Malaysia. It states in the discussion there is a previous publication from this population, but this is not mentioned in the introduction and therefore no discussion about how this paper is different. The authors state that their findings mirror the previous study – so why was there a need to conduct this study? This lack of clear rationale needs addressing. If the results of VMI in this population are already know, this study appears to be a repetition.

Introduction

P4 2nd paragraph – authors refer to published US norms but do not include the references, these should be added

Need to include previous findings specific to this population, which are in the discussion. If the VMI of this population is already known, what is the rationale for this study?

Methods

For the sample size calculation, it states a confidence level of 95% but doesn’t state what measure this is based on or the degree of change used in the sample size calculation.

Why is there double the number of government schools compared with private?

What was the rationale for using dichotomous variables for defining variables such as parents education level and household income, by only having two options there is a lack of sensitivity when evaluating the results.

Results

The data in table one are clearly presented so do not need repeating in the text.

The same applies to the gender data from table two.

Discussion

The first point made that the older age group that has the better VMI scores may be attributable to their early enrolment into preschool – did this subgroup have higher rates of children attending preschool? This analysis could be done and then a definite conclusion could be made regarding this. In the same paragraph the authors make reference to this where they have looked at the rates of children attending preschool at an earlier age across the age groups, but this data hasn’t been presented.

6. PLOS authors have the option to publish the peer review history of their article (what does this mean?). If published, this will include your full peer review and any attached files.

Reviewer #1: **Yes: **Ewa Niechwiej-Szwedo

Reviewer #2: No

---

## [Author Response · Author response to Decision Letter 0]

7 Jan 2021

Manuscript ID: PONE – D – 20 – 32756

Title: Association between reduced visual-motor integration performance and socioeconomic factors among preschool children in Malaysia: a cross-sectional study 

Editor’s Comments:

1. The authors should clearly state how this study differs from the previous literature in the same population. It appears that the authors are trying to make the case that their study is evaluating more socioeconomic factors than previous studies. If so, this needs to be explicitly stated and should be the primary goal of their study rather than the comparison to US norms which has already been done in a previous study. 

Our response:

Our study differs from the two earlier Malaysian studies (Samsudin, Abdullah, & Mat, 2015; Tahir & Yunus, 2018) in two aspects. Although these earlier studies have also measured the VMI performance of Malaysian preschoolers, Samsudin et al. only looked at the effects of age on VMI performance, while Tahir & Yunus looked at the effects of the preschools’ language of instructions on VMI performance. None of these studies have actually reported any associations between these preschoolers’ socioeconomic status and their VMI performance or compared the results statistically with the US norms. 

We agree with the Editor’s comments that the investigations of the effects of various socioeconomic factors on VMI performance should be our study's primary goal. Thus, we have made the necessary changes in the last paragraph of the Introduction (page 6, last paragraph, line 104):

“The study’s primary aim was to evaluate the influence of various socioeconomic factors on Malaysian preschoolers’ VMI performance. As VMI performance can be culture-specific (Lim et al., 2015; Ng, Chui, Lin, Fong, & Chan, 2015), we also compared these children’s VMI performance with the standardized VMI scores published in the United States (Beery & Beery, 2010).” 

2. The authors switch between the 5th and 6th edition of the Beery-VMI test (fifth in training, but 6th for testing; 5th for scoring the drawing;6th to standardize scores....) Is this a mistake? If not, please provide rationale for using different editions for various aspects of the study. How much do they differ? 

Our response:

We thank the Editor for pointing this out. This inconsistency was a mistake; the version used for the Beery-VMI test was the 6th edition. We have made the necessary changes throughout the manuscript. 

3. Why was the father's occupation not taken into account, but the mother's was? 

Our response:

All participants in our study had fathers who were formally employed. Therefore paternal employment was not taken into account in our analysis in the earlier version of the manuscript. We thank the Editor for pointing this out. In the revised version of the manuscript, we have included these findings in Results (Table 1, page 11 and Table 5, page 15) and added an explanation in Discussion on page 22, last paragraph, line 411: 

“In addition, all of the children involved in this study had fathers who were formally employed. Therefore, paternal employment status was not included as a socioeconomic variable in our analysis. This finding is unsurprising, as Malaysian societies are still, at large, divide their roles based on gender, where men are primarily expected to provide for the family economically. In comparison, women are more likely to be out of employment due to childcare duties (Hossain, 2014).” 

4. When comparing scores between different ethnic groups, and with the 2 month groups (table 3), did the others control for multiple comparisons (i.e. bonferroni correct?).

Our response: 

The multiple comparisons of scores between ethnic groups were corrected using the Tukey HSD test, which was mentioned in the submitted manuscript (page 12, first paragraph, line 218). For the 2-month age groups, we compared the mean VMI performance of each age group with the US VMI norm score (mean: 100±15) using one-sample t-test. Therefore, corrections were not needed since no multiple comparisons were actually made for the age group data. 

Journal Requirements

1. Please ensure that the manuscript meets PLOS ONE’s style requirements: 

Our response: 

We have made the necessary changes, and the revised manuscript now has followed PLOS ONE’s style requirements. 

2. Availability of questionnaires: 

Our response:

The questionnaire used in this study has been included as Supporting Information. Two documents have been uploaded: the questionnaire in its original language (Malay) and the English-translated version. 

3. Corrections for multiple comparisons: 

Our response:

We can now clarify that we have corrected for multiple comparisons in our statistical methods, as mentioned above.

4. Please state whether you accounted for clustering by school: 

Our response:

The study population was clustered by the two types of preschool, i.e., KEMAS (government) and private. In the chosen preschools, all children who gave consent went through the vision screening process. Those who fulfilled the inclusion criteria were included as participants and had their VMI performance measured. Therefore, the clustered sampling was not multi-level sampling. To make it clearer, we have clarified this on page 7, the last paragraph, line 126: 

“The study population was clustered by the type of preschool attended. The preschools were (i) KEMAS preschools that catered for children from suburban and rural areas with families of very low income (Mustafa & Azman, 2013), and (ii) private, paid-for-profit preschools. A list of KEMAS preschools obtained from the Selangor and Kuala Lumpur Community Development Departments. A list of registered private preschools was obtained from the Ministry of Education website. Preschools were selected from these lists using simple random sampling. In 2018, approximately 55% of children attended Government-run preschools while the rest attended private preschools (Ministry of Education Malaysia, 2018). In order to obtain a sample that represented the population, similar percentages of children were sampled from both types of preschools.” 

5. Avoiding statements implying causality from correlational research: 

Our response:

We thank the Editor for the suggestions. Where relevant, we have used the terms “associated with” or “associations.”

6. Data availability: 

Our response:

We can now confirm that the data can be accessed publicly. The data file has been uploaded as Supporting Document. 

Reviewers’ Comments

Reviewer #1 

Reviewer #1 found the manuscript to be well written and provided broader insights into the role of socioeconomic status in children’s development. We thank Reviewer #1 for her comments and positive feedbacks. All comments have been addressed below:

1. Abstract: provide the age range, and consider changing months to years (also in the methods section)

Our response:

In the Abstract, we have now included the age range of our participants. We have also provided their age range. Line 26 of the Abstract now reads: 

“A cross-sectional study was conducted among 435 preschoolers (mean age: 5.95±0.47 years; age range: 5.08-6.83 years)”

Relevant changes have also been made in the Results (page 11, first paragraph, line 206).

2. Intro, pg 4 (last paragraph): Children who were enrolled…. In the same paragraph the authors should add information about the effect size that was found in the reviewed studies. For example, what was the % difference between groups or the size of correlation that was mentioned?

Our response:

These information has been included in the said section. The section of the paragraph (page 4, last paragraph, line 71) now reads: 

“For example, children who enrolled into preschool at an earlier age had better reading and mathematics skills than children who joined later, with an effect size between 0.15 to 0.30 (Magnuson, Meyers, Ruhm, & Waldfogel, 2004). In addition, there was a statistically significant positive correlation between maternal education and young children’s reading and mathematics skills (r = 0.44) (Hortaçsu, 1995). Socioeconomic status also correlates significantly with young children's fine-motor performance, where a significant positive correlation between VMI performance and socioeconomic status (r = 0.44) has been reported (Dunn, Loxton, & Naidoo, 2006). A longitudinal study found that South African children with low socioeconomic background had significantly lower VMI scores than their peers from a higher socioeconomic background. When the groups were re-assessed three years later, the differences in their VMI scores persisted (Coetzee, Pienaar, & Van Wyk, 2019). 

3. Pg 5 (last paragraph): “as a whole, Beery VMI scores …” – are the authors referring to the composite (total) score?

Our response:

The paragraph's original sentence refers to the mean of the VMI standard scores for Tahir & Yunus (2018)’s participants. 

We have rephrased this part (page 5, last paragraph, line 97) to improve its clarity:

“The mean Beery-VMI scores for their 359 participants appeared similar to the US normative sample, but these results were not statistically analyzed.” 

4. Methods: the age range reported here is 5 to 6.9 yrs, are all these children starting grade 1 at the same time? Ie, are some children starting at 6 and others at 7 years? 

Our response:

In Malaysia, children typically start primary school (i.e., grade 1) the year they turn seven. Since 1998, 6-year-old children may start primary schooling at the request of their parents. Therefore, some children in Malaysia may start primary school at six and others at seven years. 

5. It’s unclear what effects were used to calculate the sample size. The authors mention that they need 380 children to detect something with a confidence level of 95%, but what is the effect they were trying to detect?

Our response:

The original reporting on our sample size calculation was insufficiently described, and we thank both Reviewers for pointing this out. The sentence has been rephrased on page 7, paragraph 2, line 119, which now reads: 

“Krejcie and Morgan (1970) formula, with the desired confidence level of 95% and precision of ±5, was used to calculate a sample size of 380. The dropout rate was estimated at 10%; therefore, the final sample size required was 418.” 

6. Pg 16 (first paragraph): “Table 6 shows that the odds…”. “model was adjusted for gender…” 

Our response:

We again thank the reviewer for the corrections. We have made the necessary corrections on page 15, the last paragraph, line 264. 

Reviewer #2: 

We thank Reviewer #2 for the comments, who also found that the paper is well-written with clearly identified objectives and appropriate analysis. Below are our responses to the comments.

1. It states in the discussion there is a previous publication from this population, but this is not mentioned in the introduction and therefore no discussion about how this paper is different. The authors state that their findings mirror the previous study – so why was there a need to conduct this study? This lack of clear rationale needs addressing. If the results of VMI in this population are already know, this study appears to be a repetition. 

Our response:

Two earlier studies conducted on the Malaysian preschool population were mentioned in the Introduction (Samsudin et al., 2015; Tahir & Yunus, 2018). These earlier studies mainly looked at the effect of age and the preschools’ language of instruction on their participants' VMI performance. None of these studies have actually reported any associations between these preschoolers’ socioeconomic status and their VMI performance or compared the results statistically with the US norms. 

Our study's main rationale was to investigate the influence of various socioeconomic factors that may influence VMI performance in Malaysian preschool children. 

We have further improved the clarity of the writing by creating a separate paragraph, adding these points in the Introduction, page 5, last paragraph, line 93:

“Samsudin et al. (2015) measured VMI performance in children attending public preschools in the north-eastern state of Kelantan. For their relatively small sample size of 4-, 5-, and 6-year-olds (n=30 for each age group), the VMI age score was up to 2.28 years lower than the children’s chronological age. Another study measured VMI performance for preschool children residing in the southern state of Melaka (Tahir & Yunus, 2018). The mean Beery-VMI scores for their 359 participants appeared similar to the US normative sample, but these results were not statistically analyzed. They also found that children attending Chinese-stream preschools had significantly higher VMI scores than those attending Malay-stream and Tamil-stream preschools. These findings suggest that VMI performance in young children may be associated with their demographic and cultural backgrounds. Why there are different VMI performance in children residing at different states in Malaysia is currently unknown.” 

We have also discussed our findings, in comparison with those of Samsudin and Tahir, in the Discussion on page 20, in the last paragraph, line 363: 

“In an earlier study, Malaysian children from the north-eastern state of Kelantan, a state with the lowest gross domestic product (GDP) per capita in Malaysia (Department of Statistics Malaysia, 2017) had reduced VMI performance (Samsudin et al., 2015). Meanwhile, children from the southern state of Melaka, a state with GDP per capita 3.5 times higher than that for Kelantan, had VMI performance comparable to the US norms (Tahir & Yunus, 2018). Melaka also has a gross household income that is 1.7 times higher than that of Kelantan (Jabatan Statistik Malaysia, 2020). The economic differences at the state and household levels may partly explain the different VMI performance of children in the two states.” 

2. P4 2nd paragraph – authors refer to published US norms but do not include the references, these should be added

Our response:

The reference where the US norms were obtained has now been included in the text on page 4, paragraph 2, line 61. 

3. Need to include previous findings specific to this population, which are in the discussion. If the VMI of this population is already known, what is the rationale for this study?

Our response:

The previous findings specific to Malaysian preschoolers have been mentioned in the Introduction (page 5, last paragraph).

The rationale for this study has been explained similar to our response to the Editor. Briefly, our study's main objective is to investigate the associations between VMI performance in Malaysian preschool children with various aspects of their socioeconomic status. Besides, we also compared the VMI performance of Malaysian preschoolers to the US norms. Although VMI has been measured in Malaysian preschoolers previously, the results have never been compared statistically with the US norms. 

4. Methods

For the sample size calculation, it states a confidence level of 95% but doesn’t state what measure this is based on or the degree of change used in the sample size calculation.

Our response:

Like our response for Reviewer 1, we have rectified the sample size calculation in Methods, which can be seen on page 7, paragraph 2, line 119. 

5. Why is there double the number of government schools compared with private?

Our response:

The study population was clustered based on the type of preschool that they attended. In 2018, approximately 55% of children attended Government-run preschools while the rest attended private preschools. Participants of this study were sampled in similar percentages to obtain a sample that represented the population, achieved at the end of data collection with ten government preschools and five private preschools. 

6. What was the rationale for using dichotomous variables for defining variables such as parents education level and household income, by only having two options there is a lack of sensitivity when evaluating the results.

Our response: 

Based on literature and local context, we took careful considerations when deciding that the outcomes should be divided into two categories. In Malaysia, the low household income group has been getting attention lately from the country’s policymakers, where various programs and aid had been planned for this group. Therefore it was decided that the variable should be divided into low- and higher-income groups. For parental education, less-educated workers in the country are defined as employees with secondary school qualifications or lower (Yunus, Bakar, Hamzah, & Jaafar, 2016). Therefore, we decided that parental education should be categorized into secondary school education and higher educational attainment. Previous studies on motor performance in young children have also used a similar dichotomy when defining their socioeconomic variables (Coetzee et al., 2019; Coetzee, Pienaar, & van Wyk, 2020; Magnuson et al., 2004). Low parental education levels and parental employment status are also associated with children's well-being and educational development around the world (Burhan, Yunus, Tovar, & Burhan, 2017). Based on these reasons, the socioeconomic variables measured in this study were mostly dichotomous. 

We agree with Reviewer 2 that having only two options would lead to a lack of sensitivity when evaluating the results. Therefore, in a future study, these variables could be studied in more detail to provide a clearer picture of their association with VMI performance in young children. We have included this as part of the study’s limitations on page 23, paragraph 1, line 424: 

“Furthermore, all of the socioeconomic variables, including household income and parental employment status, were dichotomous. Although the country's context supported the binary categorization of these variables, the dichotomy would have led to reduced statistical power to detect relationships between the variables.” 

7. Results

The data in table one are clearly presented so do not need repeating in the text.

The same applies to the gender data from table two.

Our response:

We thank Reviewer #2 for pointing this out. Indeed, it was redundant to repeat the data from Tables 1 and 2 in the text. Therefore, they have been omitted as suggested. 

8. The first point made that the older age group that has the better VMI scores may be attributable to their early enrolment into preschool – did this subgroup have higher rates of children attending preschool? This analysis could be done and then a definite conclusion could be made regarding this. In the same paragraph the authors make reference to this where they have looked at the rates of children attending preschool at an earlier age across the age groups, but this data hasn’t been presented.

Our response:

In the manuscript, we actually stated that older children tend to have lower VMI scores (page 18, line 292). For children who started preschool at the age of 6, i.e., the older children, 77.1% of them enrolled in KEMAS preschools, and 77.5% were from the low household income group. In contrast, more than half of the younger children in our group enrolled in private preschools (53.8%). Indeed, children who attend preschool programs later than their peers perform worse on various academic-related assessments (Magnuson et al., 2004). Nevertheless, the mean scores for all age subgroups were still within the Average category. Therefore, we can conclude that our sample of Malaysian preschoolers' overall Beery-VMI performance compares well with the standard US sample. 

We have rephrased this section in Discussion, page 18, paragraph 1, line 293: 

“On a closer inspection, children in the older age group were mainly from the low household income (77.5%) and enrolled in the government KEMAS preschools (77.1%). The combination of these factors in the older children could have led to late exposure to activities that involve fine-motor skills, which are part of learning in preschools, might explain why these older children tend to have lower Beery-VMI scores.” 

End of response.

Again, I would like to thank you for allowing us to address all the comments on behalf of the team. I hope that the clarifications that we have provided to be satisfactory. We look forward to a favorable Editorial response from you and the Reviewers.

---

## [Decision Letter · Decision Letter 1]

27 Jan 2021

Association between reduced visual-motor integration performance  and socioeconomic factors among preschool children in Malaysia: a cross-sectional study

PONE-D-20-32756R1

Dear Dr. Dr. Hairol,

We’re pleased to inform you that your manuscript has been judged scientifically suitable for publication following a few minor revisions (see Additional Editor Comments below) and will be formally accepted for publication once it meets all outstanding technical requirements.

Kind regards,

Krista Kelly, Ph.D.

Academic Editor

PLOS ONE

Additional Editor Comments (optional):

Please address these minor comments.

1. The sample size calculation still does not provide any information on the effect that was to be expected, i.e. how much of a difference in the VMI score between groups would be detected?

2. Line 151 A list of KEMAS preschools ‘was’ obtained….

3. Line 264. Table 6 shows that the odds ratio… remove ‘that’

4. The authors gave a reasonable response to Reviewer 2 regrading the rationale for dichotomizing variables, but this was not added to the Methods. Could the authors please add this to the Methods?

Reviewers' comments:

Reviewer's Responses to Questions

**Comments to the Author**

1. If the authors have adequately addressed your comments raised in a previous round of review and you feel that this manuscript is now acceptable for publication, you may indicate that here to bypass the “Comments to the Author” section, enter your conflict of interest statement in the “Confidential to Editor” section, and submit your "Accept" recommendation.

Reviewer #1: All comments have been addressed

Reviewer #2: All comments have been addressed

2. Is the manuscript technically sound, and do the data support the conclusions?

Reviewer #1: Yes

Reviewer #2: (No Response)

3. Has the statistical analysis been performed appropriately and rigorously? 

Reviewer #1: Yes

Reviewer #2: (No Response)

4. Have the authors made all data underlying the findings in their manuscript fully available?

Reviewer #1: Yes

Reviewer #2: (No Response)

5. Is the manuscript presented in an intelligible fashion and written in standard English?

Reviewer #1: Yes

Reviewer #2: (No Response)

6. Review Comments to the Author

Reviewer #1: (No Response)

Reviewer #2: (No Response)

7. PLOS authors have the option to publish the peer review history of their article (what does this mean?). If published, this will include your full peer review and any attached files.

Reviewer #1: No

Reviewer #2: **Yes: **Dr Anna R O'Connor

---

## [Editor Report · Acceptance letter]

9 Feb 2021

PONE-D-20-32756R1 

Association between reduced visual-motor integration performance and socioeconomic factors among preschool children in Malaysia: a cross-sectional study 

Dear Dr. Hairol:

I'm pleased to inform you that your manuscript has been deemed suitable for publication in PLOS ONE. Congratulations! Your manuscript is now with our production department. 

Kind regards, 

on behalf of

Dr. Krista Kelly 

Academic Editor

PLOS ONE